# Screening of New Potential Probiotics Strains against *Photobacterium damselae* Subsp. *piscicida* for Marine Aquaculture

**DOI:** 10.3390/ani11072029

**Published:** 2021-07-07

**Authors:** Ana Gutiérrez Falcón, Daniel Padilla, Fernando Real, María José Ramos Sosa, Begoña Acosta-Hernández, Andrés Sánchez Henao, Natalia García-Álvarez, Inmaculada Rosario Medina, Freddy Silva Sergent, Soraya Déniz, José Luís Martín-Barrasa

**Affiliations:** 1University Institute of Animal Health and Food Safety (IUSA), University of Las Palmas de Gran Canaria, 35413 Arucas, Spain; ana.gutierrez106@alu.ulpgc.es (A.G.F.); fernando.real@ulpgc.es (F.R.); maria.ramos146@alu.ulpgc.es (M.J.R.S.); bego.acosta@ulpgc.es (B.A.-H.); juansahe21@hotmail.com (A.S.H.); natalia.garcia@ulpgc.es (N.G.-Á.); inmaculada.rosario@ulpgc.es (I.R.M.); freddy.silva101@alu.ulpgc.es (F.S.S.); soraya.deniz@ulpgc.es (S.D.); joseluis.martin@ulpgc.es (J.L.M.-B.); 2Research Unit, Hospital Universitario de Gran Canaria, Dr Negrín, 35019 Las Palmas de Gran Canaria, Spain

**Keywords:** marine aquaculture, probiotics, pasteurellosis

## Abstract

**Simple Summary:**

Marine aquaculture has been one of the fastest growing animal production sectors in the last thirty years worldwide. On intensive farms, 10% of the population dies exclusively from pathogen activity, and *Photobacterium damselae* subsp. *piscicida* is one of the major infectious agents causing marine fish mortality. The purpose of this study is to obtain potential probiotic strains against pasteurellosis, in order to limit the use of chemotherapy, avoiding the environmental impact generated by the abusive use of these products. Bacterial strains isolated from different fish species were characterized in vitro and in vivo to determine their probiotic properties. Of the total number of strains isolated, only one showed excellent results to continue its characterization in vivo against this marine pathogen. The use of this strain as a possible probiotic for marine aquaculture will be in progress in future studies.

**Abstract:**

On intensive fish farms, 10% of the population dies exclusively from pathogens, and *Photobacterium damselae* subsp. *Piscicida (Ph. damselae* subsp. *Piscicida)*, the bacteria causing pasteurellosis in marine aquaculture, is one of the major pathogens involved. The objective of this study was to obtain new probiotic strains against pasteurellosis in order to limit the use of chemotherapy, avoiding the environmental repercussions generated by the abusive use of these products. In this study, 122 strains were isolated from the gills and intestines of different marine fish species and were later evaluated in vitro to demonstrate the production of antagonistic effects, the production of antibacterial substances, adhesion and growth to mucus, resistance to bile and resistance to pH gradients, as well as its harmlessness and the dynamic of expression of immune-related genes by real-time PCR after administration of the potential probiotic in the fish diet. Only 1/122 strains showed excellent results to be considered as a potential probiotic strain and continue its characterization against *Ph. damselae* subsp. *piscicida* to determine its protective effect and elucidating in future studies its use as a possible probiotic strain for marine aquaculture.

## 1. Introduction

The success of modern aquaculture is based on controlling the reproduction of used species, the best knowledge of their biology, disease control, technological innovations, and the development of specific foods. The production costs control system has become a key problem for the economic viability of this industry. On intensive farms, 10% of the population dies exclusively from pathogens, producing a huge economic loss [1,2]. Diseases occur due to unfavourable culture conditions for fish health, and this is often presented as a limiting factor that can determine the profitability of companies.

Antibiotics have not always been used responsibly in aquaculture, which has led to the development of resistant bacteria [3], in addition to causing possible risks to public health. The application of probiotics in aquaculture is related to biological control against infectious diseases, survival, increased growth and enzymatic activity, improved immune response and stress and water quality [4].

*Photobacterium damselae* subsp. *piscicida*, the causative agent of a septicaemia affecting warm water marine fish species, is one of the most important infectious diseases in aquaculture. For its control, broad-spectrum antibiotics such as oxytetracycline are used. In addition, there are several commercial vaccines with an efficacy that depends largely on the species and size of the fish [5]. The negative effects on environmental and public health caused by the indiscriminate use of antibiotics forced the European Union to restrict the use of antibiotics in aquaculture, making it necessary to develop new safe and efficient strategies to control infectious diseases, such as new vaccines, immunostimulants and probiotics [6,7,8,9].

The use of probiotics is a good alternative to antibiotics for controlling infectious fish diseases [10]. Most probiotics in aquaculture are strains of the genus *Bacillus*, *Vibrio* and *Pseudomonas* [11,12], but few probiotics are commercially available [12]. Before being considered as a probiotic strain, a strain must have a number of characteristics. It must inhibit pathogenic strains, be able to colonize the intestinal tract, and have the ability to produce immune modulation [2,6]. Potential probiotics should provide protection through the creation of a hostile environment for pathogens by the production of inhibitory compounds and by competing for adhesion sites [10]. The use of probiotics is regarded as a very promising strategy and shows a wide acceptance to be used in aquaculture [13]. In the present work, potential probiotic strains against *Ph. damselae* subsp. *piscicida* have been studied and characterized for marine fish, including the modulation of cytokine production in seabass after adding a selected probiotic strain in the diet.

## 2. Materials and Methods

Experimental procedures with fish used in this study fulfils the requirements contained in the Directive 2010/63/EU of the European Parliament and of the Council of 22 September 2010, the Spanish Government and the University of Las Palmas de Gran Canaria (Spain) guidelines for the use of laboratory animals.

### 2.1. Sampling and Isolation of Potential Probiotic Strains

A group of 12 European seabass (*Dicentrarchus labrax*) and 19 meagre (*Argyrosomus regius*) were slaughtered by immersion in anesthetic solution with clove oil. One gram of intestinal contents of each fish was homogenized in phosphate buffered saline (PBS), and serial dilutions (1/10 to 1/1000) were spread on Brain Heart Infusion Agar (BHIA, Pronadisa, Laboratories Conda, Spain) Madrid), Marine Agar (MA, Pronadisa, laboratories Conda, Spain) and Blood Agar (BA, Pronadisa, Laboratories Conda, Spain) for 48 h at 25 °C. Subsequently, gill samples were taken using a seeding handle and spread in Trypticase soy agar (TSA, Pronadisa, Laboratories Conda, Spain) and BA.

### 2.2. Antagonistic Effect of Potential Probiotic Strains against Photobacterium damselae Subsp. piscicida

This assay determines the ability of a strain isolated from fish to inhibit the growth of pathogenic strains of *Ph. damselae* subsp. *piscicida* (Table 1) by the antagonistic effect of strains to each other, probiotic and pathogen, in BHIA and BA, following the method previously described by observation of inhibition halo [14]. Strains with antagonistic effect were identified by MALDI-TOF mass spectrometry system (Autoflex III, Bruker Daltonics GmbH) [15,16].

### 2.3. Production of Antibacterial Substances of Probiotics against Pathogens

The production of antibacterial substances by the candidate probiotic strain was analyzed by the well diffusion method [17] with modifications [18]. This assay allows us to evaluate if the inhibitory effect observed against *Ph. damselae* subsp. *piscicida* is due to extracellular substances produced by the probiotics by contacting the pathogenic strain with a supernatant, concentrated 10X by prior lyophilization, of a 24-h culture of the probiotic strain.

### 2.4. Fish Bile and pH Resistance

In these assays, we tried to demonstrate that the potentially probiotic strain found is able to resist the gastrointestinal transit of fish, determining the survival of the strain after subjecting it to a pH gradient and to the action of bile (obtained from a seabass weighing 400 g) by plate counting on TSA, following the methodology previously described by Nikoskelainen et al. [17].

### 2.5. Adhesion in Intestinal and Skin Mucus

The objective of this assay is to determine the ability of the potential probiotic strain to be adhere to the mucus of the skin and intestine from a healthy seabass of 400 g approximately, as a preliminary step to the colonization of the strain in the fish. For this purpose, we followed the methodology described by Etyemez and Balcázar [10], analyzing the adhesion capacity of the strain to polystyrene plates.

### 2.6. Growth in Intestinal and Skin Mucus

In addition to the adhesion assay to different mucus, determining the ability of a candidate strain to grow in the intestinal and skin mucus of seabass is essential to determine whether the strain is able to establish itself in the intestine and remain there. For this purpose, we determined its growth capacity using the methodology described by Olsson et al. [19].

### 2.7. Probiotic Strain Safety

Potential probiotics strains were intraperitoneally injected (0.1 mL at 10^8^ CFU/mL) into two separated groups of 15 seabass with an average body weight of 15 g to determine possible adverse effects, and a control group was injected with the same volume of PBS. Both fish groups were monitored daily for 30 days after inoculation and sacrificed with an overdose of clove oil. Fish were evaluated by histopathological and microbiological methods to determine the presence of lesions and its relationship to the inoculated strain.

### 2.8. Potential Probiotic Administration in Experimental Diet

Potential probiotic strain was incorporated into the diet of a group of seabass for the analysis of their cytokine-mediated immune response. The strain was incorporated into the diet following the methodology previously described [20]. The potential probiotic strain was cultured in BHIB for 24 h at 25 °C, and then centrifuged at 2500 g for 20 min at 4 °C. The cell pellet was washed twice and resuspended in saline solution to 10^10^ CFU/mL by plate count on TSA. Then, 25 mL of the selected strain were spread on 120 g of commercial feed (Skretting, Cojobar, Spain), mixed and dried for 24 h at 37 °C, to obtain a final concentration of 10^9^ CFU per gram of the commercial feed.

Daily analysis of the feed was carried out for 3 weeks to confirm the viability of the strain in the fish pellet. For challenge, 75 seabass were fed daily with 2% of body weight for 30 days, with the experimental diet including the probiotic strain, and then the fish were slaughtered by anesthetic overdose by the analysis of gene expression. The challenge by adding the probiotic strain to the feed was made in triplicate with 25 fish per tank. Sampling was carried out at the end of feeding the fish with the experimental diet at 30 days. The control group consisted of 25 fish and was fed for 30 days with the commercial diet without the added probiotic.

### 2.9. Analysis of Gene Expression

Total RNA was extracted from liver, kidney and spleen of experimental sea bream using the Aurum™ total RNA mini kit (Biorad, Irvine, CA, USA) and finally quantified with a NanoDrop-1000 spectrophotometer (Biorad, Irvine, CA, USA). Samples were adjusted in RNase-free dH_2_O to the same concentration of 2 ng/mL and RNA was reverse transcribed to cDNA using the iScript Reverse Transcription Reagent kit (BioRad, Irvine, CA, USA) following the manufacturer’s instructions.

The expression of the selected immune-relevant genes Interleukin-1β (IL-1 β), IL-6, IL-10, Caspase 3 (casp-3), *Tumor Necrosis Factor* (TNF-α), Cyclooxygenase 2 (COX-2) and interferon-inducible Mx (Mx), were analyzed by real-time PCR using a SYBR Green Supermix (Biorad, Irvine, CA, USA). Specific PCR primers, concentration and reference are given in Table 2.

The real-time analysis consisted of 1 cycle of 95 °C for 5 min, 40 cycles of 95 °C for 15 s and annealing temperature for 30 s,1 cycle of 95 °C for 1 min, 1 cycle of 70 °C for 1 min, and a melting curve of 81 cycles (from 55 °C to 95 °C) for 30 s.

The relative gene expression was determined according to the delta-delta Ct method, also known as the 2 ^–ΔΔCT^ method [21], using the software LightCycler (ROCHE, San José, CA, USA) with the automatic normalization of the Ct values to the housekeeping gene β–actin.
ΔCt = Ct (gene of interest) − Ct (housekeeping gene β-actin)(1)
ΔΔCt = ΔCt (probiotic treated) − ΔCt (probiotic untreated)(2)
Ratio gene expression = 2^−ΔΔCt^(3)

**Table 2 animals-11-02029-t002:** Primer sequences and hybridization temperature.

Gene	Primer Sequences	Hybridization Temperature	Source
β-actin	Forward 5′ATGTGGATCAGCAAGCAGG-3′	57.7 °C	AJ537421 Genne runer
	Reverse 5′AGAAATGTGTGGTGTGGTCG-3′		
IL-1β	Forward 5′-ATTACCCACCACCCACTGAC-3′	57.7 °C	AJ269472 Genne runer
	Reverse 5′-TCTCTTCCACTATGCTCTCCAG-3′		
IL-6	Forward 5′-ACTTCCAAAACATGCCCTGA-3′	59.3 °C	AM490062 (Sepulcre et al., 2007) [22]
	Reverse 5′- CCGCTGGTCAGTCTAAGGAG-3′		
IL-10	Forward 5′-ACCCCGTTCGCTTGCCA-3′	59.3 °C	AM268529 (Picchietti et al., 2009) [23]
	Reverse 5′-CATCTGGTGACATCACTC-3′		
Casp-3	Forward 5’-ACGAAGCAGGTCAATCATCC-3’	59.3 °C	DQ345774 Genne runer
	Reverse 5’-GCAGTTTAAGGGTATCCAGAGC-3’		
TNF-α	Forward 5′-GCCAAGCAAACAGCAGGAC-3′	60 °C	DQ200910 Genne runer
	Reverse 5′-ACAGCGGATATGGACGGTG-3′		
COX-2	Forward 5′-AGCACTTCACCCACCAGTTC-3′	59.3 °C	AJ630649 (Sepulcre et al., 2007) [22]
	Reverse 5′-AAGCTTGCCATCCTTGAAGA-3′		
Mx	Forward 5′-GGTCAAGGAGCAGATCAAACAG-3′	57.7 °C	AM228974 Genne runer
	Reverse 5′-CTCGCATCAGGTTAGGGAATC-3′		

### 2.10. Statistical Analysis

Data were analyzed in triplicate by analysis of variance (ANOVA) using SPSS version 22.0 (SPSS, Chicago, IL, USA). Differences were considered statistically significant when *p* < 0.05.

## 3. Results

### 3.1. Antagonistic Effect of Potential Probiotic Strains

A total of 122 bacterial strains were isolated from the intestinal gut and gills of seabass and meagre sampled, but only 3 strains showed inhibitory effect against at least one of the pathogenic strains tested. Identification of these strains by the MALDI-TOF mass spectrometry system can be observed in Table 3, as well as their profiles of antagonistic effect on growth against different strains of *Ph. damselae* subsp. *piscicida*. The strain *Alcaligenes faecalis* subsp. *faecalis* -1 shows inhibitory effect against all the *Ph. damselae* subsp. *piscicida* strains tested. Conversely, the strains *Alc. faecalis* subsp. *faecalis* -2 and *Pseudomonas viridiflava* only showed antagonistic effect against a single pathogenic strain, *Ph. damselae* subsp. *piscicida* EP04.

### 3.2. Production of Antibacterial Substances

None of the 3 supernatants from potential probiotics strains tested produced a zone of inhibition by well diffusion method against the tested pathogens, allowing us to rule out that the observed antagonistic effect in the previous test is caused by the production of antimicrobial substances present in the culture supernatant of the probiotic culture.

### 3.3. Fish Bile and pH Resistance

Table 4 shows the bacterial viability to pH gradients of potential probiotic strains tested. As the pH decreases, the bacterial viability is reduced, but the strains *Alc. faecalis* subsp. *faecalis* -1 and *P. viridiflava* show values higher than 50% viability at pH 5. The strain *Alc. faecalis* subsp. *faecalis* -2 is the most sensitive to acidic pH, showing a viability of 41% at pH 5, and 4.9% at pH 3, while strain *P. viridiflava* is the most resistant to pH 3, with a viability of 11%. Strain *Alc. faecalis* subsp. *faecalis* -1 is the most resistant to the action of bile, showing a viability of 95.7% (*p* < 0.05). The viability of *P. viridiflava* is 46.6%, being *Alc. faecalis* subsp. *faecalis* -2 the most sensitive strain to the action of bile, showing a viability of 38.4%.

### 3.4. Adhesion in Intestinal and Skin Mucus

Strains *Alc. faecalis* subsp. *faecalis* -1 and *P. viridiflava* show good adherence to the skin mucus, with values close to 10%, while strain *Alc. faecalis* subsp. *faecalis* -2 only has an adherence of 3.3%. Regarding adherence to intestinal mucus, the results are similar, with strains *Alc. faecalis* subsp. *faecalis* -1 and *P. viridiflava* once again showing the highest adherence capacity with values of 8.5% and 4.4%, respectively, while strain *Alc. faecalis* subsp. *faecalis* -2 shows an adherence of 2.6%.

### 3.5. Growth in Intestinal and Skin Mucus

The potential probiotics strains analyzed have the ability to use the seabass intestinal and skin mucus as a source of nutrients, producing a statistically significant growth after 24 h of incubation with increments greater than 1 logarithm.

### 3.6. Probiotic Strain Safety

Of the 3 probiotic strains evaluated, the only completely harmless strain in seabass was *Alc. faecalis* subsp. *faecalis* -1. No mortality or damage in the internal organs were observed, and the inoculated strain was not recovered from internal organs. The challenge with the strains *Alc. faecalis* subsp. *faecalis* -2 and *P. viridiflava* produced some mortalities among the fish tested, but the inoculated strain was not recovered from internal organs and no significant histological lesions were observed.

### 3.7. Analysis of Gene Expression

*Alcaligenes faecalis* subp. *Faecalis* -1, the strain which showed the best in vitro characteristics as a potential probiotic, was the only strain totally harmless in seabass, so it was the strain selected to characterize the immunomodulatory effect in fish. Figure 1 shows the gene expression of the different genes analyzed in the trial after incorporating this strain into the seabass diet for 30 days. As we can see, there is subexpression of the gene 1L-1β in the organs sampled. Regarding the expression of the IL-6 gene, there is a high overexpression in the kidney, being 14.5 times higher than the expression observed in control fish without probiotic in the diet (*p* < 0.05). On the contrary, liver and spleen samples show subexpression of this same gene. The gene IL-10 shows a slight overexpression in liver and kidney samples (3 and 1.9 times), while subexpression of this gene is observed in the spleen. Similar results are observed in the expression of the COX-2 gene, with overexpression in the liver and kidney (3 and 4.3 times), and subexpression in the spleen. In the Mx gene, a slight overexpression in the spleen, while a slight subexpression of this gene in the liver and kidney samples are observed. Finally, the Casp-3 and TNF-α genes show a similar kinetics, with high overexpression in liver samples (*p* < 0.05) (13.2 and 6.9 times with respect to the control), while a slight overexpression in the kidney and subexpression in the spleen are observed in both genes.

## 4. Discussion

The intensification in the production to satisfy the demand for fish carries negative repercussions such as environmental degradation and the occurrence of infectious-contagious diseases, which are a major obstacle to aquaculture sustainability and have been increasing since the beginning of this practice [24].

To solve the problems regarding infectious disease outbreaks, antibiotics and chemotherapeutics have long been used as preventive and control measures [25]. However, an excessive and improper use of them causes adverse effects on fish and their environment [26], not only producing environmental contamination, but also the emergence of antibiotic-resistant strains [27], which can affect and negatively impact human health [28].

For these reasons, a series of limitations and prohibitions on their use were established and the search for new alternatives, or biologically effective and environmentally friendly tools to control infectious diseases began [29], such as the development of vaccines, the use of immunostimulants, or through the administration of live microorganisms that present antagonistic effect against different aquaculture pathogens. Today, such probiotics [30,31] constitute one of the most accepted control and prevention strategies in the aquaculture sector.

Gram-positive bacteria constitute the predominant group of probiotics in aquaculture, mainly lactic acid bacteria of the genus *Bacillus*. However, some genera of Gram-negative bacteria have also been used in aquaculture, such as *Aeromonas*, *Vibrio*, *Pseudomonas,* and some genera of the family *Enterobacteriaceae* [32]. In the present work, two fish species of marine aquaculture have been included, seabass and meagre, with the aim of isolating and selecting as many bacterial strains as possible with potential probiotic activity as possible. Initial pre-selection was carried out by analyzing the capacity to produce an inhibitory effect on the growth of different virulent strains of *Ph. damselae* subsp. *piscicida*, a major pathogen in marine aquaculture.

After sampling the gills and intestinal contents of the selected fish, 122 strains were isolated, but only 3 strains showed inhibitory effect against at least one virulent strain tested, highlighting *Alc. faecalis* subp. *Faecalis* -1, which produced antagonistic effect against 5 strains of *Ph. damselae* subsp. *piscicida* analyzed. But none of these 3 potential probiotics strains tested produced antimicrobial substances in its metabolism that could explain its antagonistic effect. Therefore, the inhibitory effect observed in the previous test may be due to the presence of volatile substances [11].

One of the fundamental requirements for selecting a strain as a probiotic candidate is that it must be able to survive acidic pH and bile action, so that it can resist gastrointestinal transit and colonize the intestine when administered orally to fish [20,33,34]. Bile extracted from seabass does not affect the viability of *Alc. faecalis* subsp. *faecalis* -1, but it does significantly affect (*p* < 0.05) the other two potential probiotic strains. However, it must be considered that the concentration of bile used in this assay (10%) is very high compared to the physiological concentration in humans, which is approximately 3%. Since the real concentration of bile in fish is not yet known, to carry out this assay, an overestimation of the concentration of bile salts was performed [17].

Low pH values affected the viability of all strains tested. However, it should be considered that the administration of probiotic strains is generally administered orally along with the feed, so the action of the stomach secretions is not direct, as the bacteria is covered within the food bolus [9,17]. In addition, other elements such as water, inorganic ions, as well as the number of daily feed intakes are involved in the digestive process [35]. *Pseudomonas viridiflava* is the most resistant strain in acidic media, but we must consider that if the probiotic strain was to be used in larvae, there should be no problem with acid pH because, in early stages of fish development, the stomach pH is moderately alkaline [36], the larval digestive tract is not yet fully developed, and bile is not secreted until later in development [37].

At this point, high survival capacity showed by the strain *Alc. faecalis* subp. *Faecalis* -1 in acidic pH and bile, stands out from the other potential probiotic strains, with a survival rate of 74.3% at pH 5, and 95.7% in contact with bile. These data, together with its enormous versatility in inhibiting in vitro the growth of all the pathogenic strains analyzed in this study, make it an ideal candidate for further characterization in vitro and subsequently in vivo, and they also bring it closer to its definitive proposal as a probiotic strain in marine aquaculture.

The ability of adhesion to mucus is an indispensable quality that every probiotic must have, since it constitutes the first step for the strain to establish itself in the intestine and colonize it [38], becoming part of the first defensive barrier of fish. Otherwise, the probiotic strain would not be able to survive and compete with pathogens, so its beneficial effects on the host would be transient [37,39]. The results obtained in the mucus adhesion assay were homogeneous, with the strains *Alc. faecalis* subp. *Faecalis* -1 and *P. viridiflava* showing the highest adhesion capacity in the two types of mucus tested. Once it has shown that the probiotic strain adheres to the intestinal epithelium, the next step would be to grow in the mucus to become established in the intestine [38]. All strains were able to grow in the intestinal mucus of seabass, and also in the cutaneous mucus after 24 h of incubation [38], showing an increasing concentration by at least one logarithm, which indicates that our potentially probiotic strains will be able to establish in the intestinal and remain there.

*Alcaligenes faecalis* subp. *Faecalis* -1 was the only strain completely harmless for seabass, so it was the strain selected to characterize its immunomodulatory effect in seabass after adding it to a commercial feed for a period of 30 days. Immunomodulation is one of the most studied mechanisms in probiotics [37,39,40,41]. The literature lists a large number of probiotic strains capable of stimulating both innate and specific immune systems in aquatic organisms [42], including modulation of cytokine production [32,43]. Thus, it has been observed that different strains of the genus *Lactobacillus*, *Bacillus subtilis* and *Enterococcus faecium*, modulate the production of proinflammatory cytokines such as IL-1β, TNF-α, and IL-6, and anti-inflammatory cytokines such as IL-10, IL-8 and transforming growth factor-β (TFG-β), suggesting that probiotic-fed fish are in an immunologically elevated state to counter any possible infections by activating a variety of pro and anti-inflammatories mediators [44,45]. In this study, we determined the expression of cytokines that are directly related to the immune response such as IL-1β, IL-6, IL-10, TNF-α, COX-2, as well as the Mx gene and Casp-3 by qPCR from different organs in juvenile seabass.

In this assay, we can observe that the probiotic strain induces a strong pro-inflammatory response, with strong peak with significant differences (*p* < 0.05) in IL-6 and TNF-α (*p* < 0.05), while COX-2 showed a slight over-expression, but no statistical differences were observed. However, the proinflammatory cytokine IL-1β is under-expressed. Except for the subexpression detected for 1L-1β, our results are similar to those obtained by other authors. Thus, overexpression of proinflammatory cytokines such as TNF-α and IL-6 in olive flounder (*Paralichthys olivaceus*) has been observed after administration of probiotics of the genus *Bacillus* spp. [46], and in head-kidney leucocytes of seabass, the probiotic *Vagococcus fluvialis* induced a strong pro-inflammatory response, at very high levels of expression for IL-1 β, TNF-α, IL-6 and COX-2 [47].

COX-2 is considered as a central mediator during inflammation and its overexpression is associated in vivo with chronic diseases [48], and it was slightly up-regulated by the probiotic in the present work. In contrast, it has been found that transcripts significantly lower cytokine in seabass larvae fed with *Lactobacillus delbrueckii* [23]. These cytokines are believed to contribute to the host defence mechanisms in response to the colonization and invasion.

We have also detected a slight statistically non-significant increase in the expression of the anti-inflammatory cytokine IL-10. This cytokine down-regulates the expression of other cytokines, mainly TNF-α [49], and one of the main functions appears to be related to the regulation of response; to prevent the inflammatory response is excessive [50]. The overexpression of IL-10 has been previously observed by other authors with the probiotic *Lactococcus lactis* against *Aeromonas hydrophila* [51] and *Bacillus velezensis JW* in goldfish (*Carassius auratus*) [52].

Caspase-3 is the responsible cytokine for shutting down the inflammatory response, and it is involved in biological processes as the apoptosis, playing an important role in the homeostasis and regulation of the host response during infection [53]. In our assay, the expression values of this cytokine were highly variable among samples, observing a statistically significant overexpression in the liver, while in the kidney we observed a slight non-significant overexpression and subexpression in the spleen.

Finally, Mx proteins are the molecular effector of interferon, one of the most known mechanisms against virus [54]. Mx gene can be stimulated by bacteria in fish, as different authors have shown [55,56]. No statistically significant differences were found with respect to the control, showing a very slight expression of the Mx gene in the spleen, while in the liver and kidney, subexpression of this gene was observed. Consequently, we cannot consider that this potential probiotic strain is capable of activating the interferon system in seabass.

This study has demonstrated that the oral administration for 30 days of *Alc. faecalis* subp. *Faecalis* -1 can stimulate the immune response in fish by activating a variety of pro and anti-inflammatory mediators and, therefore, can be an important regulator of seabass gut-associated immune system, raising the immunologically state to counter any possible infections by different pathogens, particularly, *Ph. damselae* subsp. *Piscicida.*

## 5. Conclusions

Of the 3 strains characterized, only *Alc. faecalis* subsp. *faecalis* -1 is a suitable candidate as a potential probiotic strain in aquaculture against *Ph. damselae* subsp. *piscicida*. In addition, this strain modulates the nonspecific immune response in seabass after administration in feed for a period of 30 days, showing different levels of expression, activating pro-inflammatory cytokines such as IL-6, COX-2 and TNF-α, as well as Casp-3.

## Figures and Tables

**Figure 1 animals-11-02029-f001:**
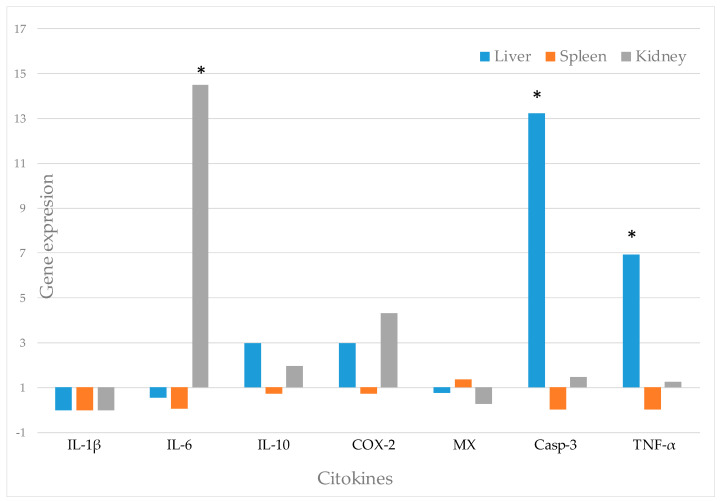
Gene expression of IL-1β, Il-6, IL-10, COX-2, Mx, Casp-3 and TNF-α in seabass after *Alc. faecalis* subsp. *faecalis* -1 administration in experimental diet for 30 days. Asterisks denote significantly different (*p* < 0.05).

**Table 1 animals-11-02029-t001:** Pathogenic strains used in testing antagonistic effect.

Pathogenic Strains	Reference	Source
*Ph. damselae* subsp. *piscicida* C2	IUSA	*Sparus aurata*
*Ph. damselae* subsp. *piscicida* 17911	ATCC	*Perca fluviatilis*
*Ph. damselae* subsp. *piscicida* DI21	ATCC	*Sparus aurata*
*Ph. damselae* subsp. *piscicida* 94/99	IUSA	*Sparus aurata*
*Ph. damselae* subsp. *piscicida* EP04	IUSA	*Sparus aurata*

ATCC: American Type Culture Collection. IUSA: Instituto Universitario de Sanidad Animal y Seguridad Alimentaria.

**Table 3 animals-11-02029-t003:** In vitro antagonistic effect and identification of potential probiotic strains isolated from seabass and meagre against different virulent strains of *Ph. damselae* subsp. *piscicida*.

Pathogenic Strains Tested	Potential Probiotic Strains		
	1	2	3
*Ph. damselae* subsp. *piscicida* C2	+	-	-
*Ph. damselae* subsp. *piscicida* ATCC 17911	+	-	-
*Ph. damselae* subsp. *piscicida* DI21	+	-	-
*Ph. damselae* subsp. *piscicida* 94/99	+	-	-
*Ph. damselae* subsp. *piscicida* EP04	+	+	+

1.- *Alcaligenes faecalis* subsp. *faecalis* -1; 2.- *Alcaligenes faecalis* subsp. *faecalis* -2; 3.- *Pseudomonas viridiflava*.

**Table 4 animals-11-02029-t004:** Survival (%) to pH gradients of potential probiotic strains tested.

pH	Potential Probiotic Strains		
	1	2	3
pH 7	100	100	100
pH 6	91.6 ± 3.3	71.8 * ± 2.2	85.7 ± 3.5
pH 5	74.3 * ± 4.5	41.5 * ± 3.5	65.5 * ± 1.2
pH 4	50.4 * ± 10.6	25.2 * ± 2.7	55.6 * ± 0.8
pH 3	6.3 * ± 3.8	4.9 * ± 1.8	11.5 * ± 0.9

Asterisks “*” denote significantly different (*p* < 0.05); 1.- *Alc. faecalis* subsp. *faecalis* -1; 2.- *Alc. faecalis* subsp. *faecalis* -2; 3.- *P. viridiflava.*

## Data Availability

The data presented in this study is contained within this article.

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
