# Peer review of "Screening of New Potential Probiotics Strains against Photobacterium damselae Subsp. piscicida for Marine Aquaculture"

_animals, 2021, doi:10.3390/ani11072029_

Round 1

Reviewer 1 Report

In this study, authors focused on probiotics to prevent  disease caused by Photobaterium damselae subsp. piscicida. Probiotics has been expected as alternative measure to drugs. This is very meaningful in sustainable aquaculture industry. However, authors should carefully revise this manuscript for publish in this journal. 

1) Authors evaluated the adhesion ability of probiotic candidates to intestinal mucus. In case of mammalian, the surface of epithelial cells in intestine  is consists of mucus layer composed by mucin. Intestinal bacteria colonize there, and form intestinal flora in the host animal. However, in case of fish, recent study showed that the surface of intestinal wall is covered by chitinous filament to avoid bacterial adhesion. So it means that the existence of intestinal bacteria in fish intestine is transient. This is quite different with the case of mammalian. Why did author evaluate the adhesion of bacteria in intestinal mucus.

2) In results, authors showed the values with the last digit, but sometimes, without the last digit. For example, 41% and 46.6%. Authors should unify this.

3) Totally, please carefully re-check and revise the detail of entire manuscript.

For example, authors notes as p <0.05 with space, but sometimes, p<0.05 without space. 

4) The content of introduction is not enough. Authors should present more information.

<Title>

・In the title of this manuscript, "damselae" is scientific name. So, this word should be italic. Please check it.

<Figures>

・In Figure 1, the title of Y-axis should be noted, but nothing. In addition, please explain the meaning of "*" in figure legend.

<Tables>

・The form of Table looks like not formal. Please check the official format to prepare the manuscript in Journal, animals.

・In Table a, "Table 1" in title is written in Italics. But, in latest issues of this journal, "Table" is not italic. Please check it. And also, the "Perca fluviatilis" is written as Bold. Is there a reason to write it in Bold?

・In Table 1, the name of strain is written as "DI21" without "-" in the name. However, in Table 3, the name is written as "D1-21". Authors should unify this.

・In Table 3, the scientific name should be italic.

・In Table 4, the meaning of "*" should be noted in the footnote.

<References>

・Please re-check the style of references.  I read latest issue in this Journal. For example, the volume should be italic.

<Others>

・I recommend to carefully re-check entire manuscript. I found many mistakes. For example, the usage of tense is confused. Please re-check  the tense, which one is adequate, "is" or "was". And also, the name of scientific name should be italic. 

Author Response

1) Authors evaluated the adhesion ability of probiotic candidates to intestinal mucus. In case of mammalian, the surface of epithelial cells in intestine  is consists of mucus layer composed by mucin. Intestinal bacteria colonize there, and form intestinal flora in the host animal. However, in case of fish, recent study showed that the surface of intestinal wall is covered by chitinous filament to avoid bacterial adhesion. So it means that the existence of intestinal bacteria in fish intestine is transient. This is quite different with the case of mammalian. Why did author evaluate the adhesion of bacteria in intestinal mucus.

I was unaware of this recent research about the limited capacity of bacterial adherence at the intestinal level of fish. This methodology has been used as it is widely described in the search for probiotics in fish, as a step prior to colonization and subsequent multiplication in fish intestines. 
Specifically, we have results from this strain (unpublished), that after several weeks of having finished the diet with the probiotic, we continue to isolate the probiotic at the intestinal level, so although intestinal adherence may be limited by these recent studies, we believe that the data are still of interest.

2) In results, authors showed the values with the last digit, but sometimes, without the last digit. For example, 41% and 46.6%. Authors should unify this.

- Revised and modified

3) Please carefully re-check and revise the detail of entire manuscript.

- Revised

4 The content of introduction is not enough. Authors should present more information.

- Text added in the introduction

5 In the title of this manuscript, "damselae" is scientific name. So, this word should be italic. Please check it.

- Modifed

6 In Figure 1, the title of Y-axis should be  noted, but nothing. In addition, please explain the meaning of "*" in figure legend.

added Y-axis and the meaning of  "*"in the figure legen

7 The form of Table looks like not formal. Please check the official format to prepare the manuscript in Journal, animals.

・In Table a, "Table 1" in title is written in Italics. But, in latest issues of this journal, "Table" is not italic. Please check it. And also, the "Perca fluviatilis" is written as Bold. Is there a reason to write it in Bold?

Modified

8 In Table 1, the name of strain is written as "DI21" without "-" in the name. However, in Table 3, the name is written as "D1-21". Authors should unify this.

Modified

9 In Table 3, the scientific name should be italic.

Modified

10 In Table 4, the meaning of "*" should be noted in the footnote.

added

11 <References>

・Please re-check the style of references.  I read latest issue in this Journal. For example, the volume should be italic.

checked

12 I recommend to carefully re-check entire manuscript. I found many mistakes. For example, the usage of tense is confused. Please re-check  the tense, which one is adequate, "is" or "was". And also, the name of scientific name should be italic. 

checked

Reviewer 2 Report

The paper entitled “Screening of new potential probiotics strains against Photobacterium damselae subsp. piscicida for marine aquaculture” describes the selection of potential probiotics strains to be applied in marine aquaculture and their in vitro characterization.

The topic is of interest and the study is well designed, however, some points should be addressed.

Under the introduction at lanes 49-53, the authors mentioned the negative effects of the use of antibiotics and the need to develop new strategies, safe and efficient, for the control of infectious diseases. I think to this point also the vaccine strategy could be introduced as a good alternative to prevent pasteurellosis and the proper references should be cited.

Under the material and methods, different assays are cited and the methodology referred to other papers: see 2.2, 2.3, 2.4, and 2.5 paragraphs. I think a brief description of the methods could be appropriate to help understand the read.

Under the 2.8 paragraph, the authors describe the challenge. Do they mean challenge with the potential probiotic strain? This should be better clarified, otherwise, I can think also about the challenge with Photobacterium piscicida subsp. piscicidae strain that is not tested in the present work.

How many fish are included in the control group? Please specify.

Two strains of Alcaligenes faecalis subsp. faecalis have been isolated and indicated along with the paper as Alc. faecalis subsp. faecalis-1 and Alc. faecalis subsp. faecalis-2. I think the strain number should be separated from the subsp. faecalis.

The 3.2 paragraph is not clear. Please describing better the method under the material and methods section and revise the results paragraph.

At lane 210, the authors report “The challenge with the strains Alc. faecalis subsp. faecalis-2 produced some mortalities among the fish tested”. How can they explain the differences in mortality between strains of the same species? Furthermore, is this species reported as probiotic?

At lanes 263-265, please discuss the reported results.

Please revise the sentence at lanes 275-278.

Minor points:

At lane 52, please correct “an efficient” with “and efficient”.

At lane 55, Bacillus in italic.

In table 4, specify the superscript *.

At lane 316, delete “directly related to immune”, repeated twice.

Author Response

Under the introduction at lanes 49-53, the authors mentioned the negative effects of the use of antibiotics and the need to develop new strategies, safe and efficient, for the control of infectious diseases. I think to this point also the vaccine strategy could be introduced as a good alternative to prevent pasteurellosis and the proper references should be cited.

Added the text "as new vaccines, immunostimulants and probiotics". References 5-8 already include the comments in the text.

Under the material and methods, different assays are cited and the methodology referred to other papers: see 2.2, 2.3, 2.4, and 2.5 paragraphs. I think a brief description of the methods could be appropriate to help understand the read.

The first version of the manuscript included a detailed description of the methodology, but the manuscript was initially rejected by the editors of the journal because according to an anti-plagiarism software, it reported that the methodology, which is what it is and I cannot change it, was plagiarized from other articles including some published by our research group. That is why we decided to modify the methodology by describing the methods in a general way. If a reader is interested in replicating the methodology, it is easily accessible through the cited references.

Under the 2.8 paragraph, the authors describe the challenge. Do they mean challenge with the potential probiotic strain? This should be better clarified, otherwise, I can think also about the challenge with Photobacterium piscicida subsp. piscicidae strain that is not tested in the present work.

Added text ...the challenge "by adding the probiotic strain to the feed"

How many fish are included in the control group? Please specify.

Added text "consisted of 25 fish"

Two strains of Alcaligenes faecalis subsp. faecalis have been isolated and indicated along with the paper as Alc. faecalis subsp. faecalis-1 and Alc. faecalis subsp. faecalis-2. I think the strain number should be separated from the subsp. faecalis.

Change Alcaligenes faecalis subsp. faecalis-1 to Alcaligenes faecalis subsp. faecalis -1

The 3.2 paragraph is not clear. Please describing better the method under the material and methods section and revise the results paragraph.

The methodology has been expanded, and the 3.2 paragraph it has been tried to make it clearer

At lane 210, the authors report “The challenge with the strains Alc. faecalis subsp. faecalis-2 produced some mortalities among the fish tested”. How can they explain the differences in mortality between strains of the same species? Furthermore, is this species reported as probiotic?

This species has not been previously described as a probiotic species. There are some descriptions that it may even be pathogenic in immunocompromised individuals, but our fish safety tests rule out any adverse effects in fish.
On the other hand, the use of species considered possible pathogens in fish are widely used as probiotics in other species, as is the case with vibrio alginolitycus in shrimp farming.

At lanes 263-265, please discuss the reported results.

Added the text "Therefore, the inhibitory effect observed in the previous test may be due to the presence of volatile substances [10]."

Please revise the sentence at lanes 275-278.

Text modified

- At lane 52, please correct “an efficient” with “and efficient”.

Done

- At lane 55, Bacillus in italic.

Done

- In table 4, specify the superscript *.

added "Asterisks denote significantly different (p < 0.05)".

- At lane 316, delete “directly related to immune”, repeated twice

   Done

Reviewer 3 Report

MDPI

Animals-1242173

In this manuscript, the authors isolated 122 strains from the gills and intestine of marine fish and evaluated in vitro to demonstrate the production of antagonistic effects, and obtained a new probiotics strain, Alcaigenes faecalis subp, faecalis-1. The authors conducted analysis of gene expression of Il-1β, IL-6, IL-10, Casp-3, TNF-α, Cox-2 and Mx. The strain appeared to modulate the nonspecific immune response in seabass. It showed excellent results to be considered as potential probiotic strain against Photobacillum damselae subsp. piscicida. This is a well written manuscript. The authors have obtained new information in the field of probiotics for marine aquaculture.

  1. Page 3, 2.4. Fish bile and pH resistance, Line 5: Change Nikoskelain [16] to Nikoskelainen et al [16].
  2. Page 5, Table 2, Source, AM490062 (Sepulcre et al., 2007), AM268529 (Picchietti et al., 2009), AJ630649 (Sepulcre et al., 2007): Suggest revise them, and add references in the Reference section.

Author Response

1 Page 3, 2.4. Fish bile and pH resistance, Line 5: Change Nikoskelain [16] to Nikoskelainen et al [16].

Done

2) Page 5, Table 2, Source, AM490062 (Sepulcre et al., 2007), AM268529 (Picchietti et al., 2009), AJ630649 (Sepulcre et al., 2007): Suggest revise them, and add references in the Reference section.

Revised and references added in reference section